# Tailoring CO_2_/CH_4_ Separation Performance of Mixed Matrix Membranes by Using ZIF-8 Particles Functionalized with Different Amine Groups

**DOI:** 10.3390/polym11122042

**Published:** 2019-12-09

**Authors:** Nadia Hartini Suhaimi, Yin Fong Yeong, Christine Wei Mann Ch’ng, Norwahyu Jusoh

**Affiliations:** 1Chemical Engineering Department, Universiti Teknologi PETRONAS, Seri Iskandar 32610, Perak, Malaysia; nadia_17005943@utp.edu.my (N.H.S.); christine_18001973@utp.edu.my (C.W.M.C.); norwahyu.jusoh@utp.edu.my (N.J.); 2CO_2_ Research Centre (CO_2_RES), R&D Building, Universiti Teknologi PETRONAS, Seri Iskandar 32610, Perak, Malaysia; 3Centre for Contaminant Control & Utilization (CenCoU), Chemical Engineering Department, Universiti Teknologi PETRONAS, Seri Iskandar 32610, Perak, Malaysia

**Keywords:** ZIF-8, Amine-Functionalized Filler, Mixed Matrix Membranes

## Abstract

CO_2_ separation from CH_4_ by using mixed matrix membranes has received great attention due to its higher separation performance compared to neat polymeric membrane. However, Robeson’s trade-off between permeability and selectivity still remains a major challenge for mixed matrix membrane in CO_2_/CH_4_ separation. In this work, we report the preparation, characterization and CO_2_/CH_4_ gas separation properties of mixed matrix membranes containing 6FDA-durene polyimide and ZIF-8 particles functionalized with different types of amine groups. The purpose of introducing amino-functional groups into the filler is to improve the interaction between the filler and polymer, thus enhancing the CO_2_ /CH_4_ separation properties. ZIF-8 were functionalized with three differents amino-functional group including 3-(Trimethoxysilyl)propylamine (APTMS), *N*-[3-(Dimethoxymethylsilyl)propyl ethylenediamine (AAPTMS) and *N*^1^-(3-Trimethoxysilylpropyl) diethylenetriamine (AEPTMS). The structural and morphology properties of the resultant membranes were characterized by using different analytical tools. Subsequently, the permeability of CO_2_ and CH_4_ gases over the resultant membranes were measured. The results showed that the membrane containing 0.5 wt% AAPTMS-functionalized ZIF-8 in 6FDA- durene polymer matrix displayed highest CO_2_ permeability of 825 Barrer and CO_2_/CH_4_ ideal selectivity of 26.2, which successfully lies on Robeson upper bound limit.

## 1. Introduction

Natural gas primarily consists of methane (CH_4_), with the small presence of other hydrocarbons including ethane (C_2_H_6_), propane (C_3_H_8_), butane (C_4_H_10_) as well as impurities consisting of carbon dioxide (CO_2_), nitrogen (N_2_), oxygen (O_2_) and hydrogen sulfide (H_2_S) [1]. These impurities, especially CO_2_, need to be separated from CH_4_ in order to meet pipeline quality standard specifications and to avoid pipeline corrosion [2]_._

Technologies used for CO_2_ removal include amine absorption [3], chemical adsorption [4], cryogenic distillation [5] and membrane technology [6]. Corrosion of the absorber column caused by the solvent used such as amines and Benfield solution are the main challenge faced by amine absorption process. Besides, the disposed used solvent could eventually lead to environmental risks. On the other hand, solvent loss, emission of VOCs and high energy requirements are the major drawbacks for chemical absorption [7]. The main concern of cryogenic distillation is the high energy requirement which increases the operation cost, although it is suitable for purification of feed gas with a high concentration of CO_2_ [8]. Among these technologies, membrane separation offers significant advantages including simple operation, low capital cost and low energy demand [9]. Thus, membrane separation technology receives great attention over the years.

Revolution of membrane separation technology starts with polymeric membranes [10], followed by inorganic membranes [11,12] and mixed matrix membranes [13,14]. Mixed matrix membranes have become a major focus because polymeric membranes suffer from trade-off between permeability and selectivity of Robeson upper bound [15], while inorganic membranes are facing high cost and reproducibility issues [16]. On the other hand, mixed matrix membranes consist of polymer matrix as continuous phase and inorganic fillers as dispersed phase [17]. Commonly used polymers in gas separation include polysulfone [18,19,20], Matrimid [21,22,23], pebax [24,25,26] and 6FDA-based polyimides [27,28,29]. Meanwhile, numerous kinds of inorganic fillers including zeolite [30,31,32], carbon materials [20,33,34] and metal organic frameworks (MOFs) [24,27,28,35,36,37] are widely reported in the literature.

Metal organic frameworks (MOFs) are composed of metal atoms or clusters linked by organic ligands [5]. Due to the characteristic of MOFs such as high surface area and porosity [38,39] and tunable chemical structure [40], researchers have focused on the application of these materials as a filler in the fabrication of mixed matrix membranes. A sub-group of MOFs known as zeolitic imidazole frameworks (ZIFs), has also been explored as a potential filler for gas separation including ZIF-8 [41,42,43,44], ZIF-11 [29] and ZIF-94 [28]. ZIF-8 fillers received more attention because of their pore aperture of 3.4 Å in diameter, which can adsorb small gas molecules such as hydrogen and carbon dioxide [41]. Jusoh et al. fabricated ZIF-8/6FDA-durene mixed matrix membranes and their results showed that the membrane loaded with 10 wt% of ZIF-8 the performance was enhanced and successfully surpassed the 2008 Robeson upper bound trade-off limit. In another work, Nordin et al. [44] reported on ZIF-8/PSf mixed matrix membranes. They found that the CO_2_ permeability and CO_2_/CH_4_ selectivity were enhanced by 37% and 19% compared to pure PSf membrane. Ordonez et al. embedded ZIF-8 in Matrimid and the results showed that permeability increased as the loading of ZIF-8 increased up to 40 wt%. However, when further increasing the loadings to 50 wt% and 60 wt% the gas permeability decreased. This might be because of the transition from a polymer-driven to a ZIF-8-controlled gas transport process [41], where at higher loadings the sieving impact of the ZIF-8 fillers is more apparent.

It has been reported from the literature that common problems faced by the fabrication of mixed matrix membranes include non-uniform distribution and agglomeration of inorganic fillers, polymer rigidication as well as incompatibility between filler and polymer. This has led to a new approach by modifying fillers with amine functional group. Over the past few years, the incorporation of amine-functionalized MOFs into polymer has [24,45,46,47,48,49] resulted in better gas separation performance compared to the membrane loaded with non-functionalized fillers. This is mainly due to the presence of amine groups in the fillers which exhibited strong affinity to acidic gas, particularly CO_2_. Amedi and Aghajani incorporated 3(trimethoxysilyl)-propylamine (APTMS) and 3-(triethoxysilyl)-propylamine (APTES) modified ZIF-8 filler into polyether block amide. They found that CO_2_ permeability increased without remarkable change on CO_2_/CH_4_ selectivity for APTES-ZIF-8/polyether block amide mixed matrix membrane [45].

Nordin et al. [46] reported the performance of NH_2_-ZIF-8/PSf mixed matrix membrane in CO_2_/CH_4_ separation. They found that the resultant membrane exhibited improvement in gas pair selectivity by 88% [46]. On the other hand, non-functionalized and amine-functionalized MIL-53 were incorporated into 6FDA-ODA polyimide by Chen et al. [47]. The resultant mixed matrix membrane showed increment in gas pair selectivity along with the increase in filler loading, mainly due to the hydrogen bonding between the filler and polymer, which has improved the interfacial properties between the polymer and filler. Meanwhile, Meshkat et al. [24] reported the performance of MIL-53 and NH_2_-MIL-53/Pebax mixed matrix membranes in CO_2_/CH_4_ separation. CO_2_ permeability and CO_2_/CH_4_ selectivity increased with the incorporation of MIL-53 and NH_2_-MIL-53 filler. These results were mainly attributed to the high porosity of filler beside the selective CO_2_ adsorption on the filler [24].

Although a number of studies have reported on the fabrication of mixed matrix membranes by using amine-functionalized filler, these studies mostly focused on monoamine group. Therefore, in this work, we functionalized ZIF-8 fillers with a different number of amine groups including monoamine, diamine and triamine and incorporated the resultant fillers into 6FDA-durene polymer matrix. The effect of functionalization with different number of amine groups towards the structural properties as well as CO_2_/CH_4_ separation performance of the resultant mixed matrix membranes was investigated.

## 2. Materials and Methods

### 2.1. Materials

In order to synthesize 6FDA-durene polyimide, 3,6-diaminodurene (durene diamine, 99% trace metal basis) and 4,4′–(hexafluoroisopropylidene) diphthalic anhydride (6FDA, 99% purity) were used. Both monomers were purified by using re-crystallization in methanol and vacuum sublimation, respectively. While propionic anhydride (PA, ≥98% purity), triethylamine (TEA, ≥99% purity), methanol (CH_3_OH, ≥99.9% purity) and dichloromethane (DCM, ≥99.8% purity) were used as received. Zinc nitrate hexahydrate (Zn(NO_3_)_2_.6H_2_O, >98% purity), 2-methylimidazole (2-MeIM, 98% purity), 3-(trimethoxysilyl)propylamine (APTMS, ≥97% purity), *N*-[3-(dimethoxymethylsilyl)propyl ethylenediamine (AAPTMS, ≥97% purity), *N*^1^-(3-trimethoxysilylpropyl) diethylenetriamine (AEPTMS) and methanol (CH_3_OH, 99.8% purity) were used without further purification for the synthesis of ZIF-8 and amine-functionalized ZIF-8. All chemicals were purchased from Sigma-Aldrich Sdn Bhd. (Kuala Lumpur, Malaysia). For gas permeation test, methane (CH_4_) and carbon dioxide (CO_2_) with 99.95% of purity were purchased from Air Product Malaysia Sdn Bhd (Kuala Lumpur, Malaysia).

### 2.2. Preparation of 6FDA-Durene Polyimide

Then, 6FDA-durene polyimide was synthesized by using two-step polycondensation reaction as reported in literature [30]. Polymerization took place between 6FDA dianhydride and durene diamines monomers to form polyamic acid (PAA). Subsequently, propionic anhydride and triethylamine were added, and polyimide solution was formed. The polyimide solution was precipitated in methanol and washed several times before being dried in vacuum oven at 150 °C for 24 h.

### 2.3. Synthesis of ZIF-8 and Amine-Functionalized ZIF-8 Fillers

ZIF-8 was prepared by following the procedure as described in the literature [50]. Firstly, 2 methyl-imidazole and zinc nitrate hexahydrate were dissolved in methanol, separately. Then, zinc nitrate hexahydrate solution was rapidly poured into 2 methyl-imidazole solution and the mixture was stirred for 1 h before centrifuged for particles recovery. The collected ZIF-8 particles were washed with methanol for few times before dried in oven at 60 °C for 24 h.

Amine-functionalized ZIF-8 was prepared according to the procedure reported by Aghajani [45]. Thus, 1 g of ZIF-8 powder was suspended in a solution containing 50 mL of desired amine-functionalized agents (APTMS, AAPTMS and AEPTMS) and 100 mL of methanol. The mixture was refluxed at 110 °C for 2 h. The amine-functionalized ZIF-8 fillers were collected by centrifugation and washed 3 times with methanol. The collected fillers were dried in a vacuum oven at 60 °C for 24 h.

### 2.4. Fabrication of Membrane

Pure 6FDA-durene membrane was fabricated by using the solvent evaporation method [51]. A 2% w/v of polymer solution was prepared by dissolved 6FDA-durene polyimide in DCM before being filtered and cast on a petri dish. Then, the resultant membrane was dried in an oven at 60 °C for 24 h and continued in vacuum conditions for another 24 h. Before it proceeded to thermal annealing at 250 °C for 24 h, the temperature was increased from 60 °C to 250 °C at a heating rate of 25 °C/hour for complete removal of residual solvent.

Mixed matrix membranes containing 0.5 and 1.0 wt% of amine-functionalized ZIF-8 fillers were fabricated based on the procedure reported in the literature [52]. The 6FDA-durene polymer was added into DCM and stirred until dissolved. Meanwhile, amine-functionalized ZIF-8 fillers were stirred in DCM and sonicated alternately to disperse fillers in DCM solution. Then, 10 wt% of 6FDA-durene polymer solution was added into amine-functionalized ZIF-8 suspension, and subsequently stirred and sonicated. After priming, the remaining of 6FDA-durene polymer solution was added into the amine-functionalized ZIF-8 suspension and the mixture was again stirred and sonicated. The mixture was stirred vigorously for 1 h before being cast onto petri dish for solvent evaporation at room temperature for 24 h. After that, the membrane was dried and heat-treats using the abovementioned procedure. Table 1 summarizes the mixed matrix membrane samples fabricated in this work.

### 2.5. Characterization of Fillers and Membranes

The crystallinity of the resultant ZIF-8 fillers was attained using X-ray diffractometer equipment (X’Pert^3^ Powder, Malvern Panalytical Ltd., Royston, UK) by using CuKa as a radiation source with a wavelength of 1.54059 Å in 2 theta range of 5° to 25°. The membrane morphology was examined by using field emission scanning electron microscope (FESEM, Zeiss Supra 55 VP, Carl Zeiss NTS GmbH, Oberkochen, Germany) operated at 10 kV under vacuum conditions. The membranes were fractured cryogenically in liquid nitrogen and sputter coated with platinum using Quorum Q150R S sputter coater (Quasi-S Sdn. Bhd., Penang, Malaysia) prior to imaging. On the other hand, the particle size of the functionalized fillers was examined via FESEM characterization (Schottky FESEM SU5000, Hitachi VP, USA) and the images are shown in Appendix A. The average particle size is about 40 nm for all the resultant fillers. The presence of Zn element in the resultant mixed matrix membranes were analyzed by using Oxford Instrument Inca energy (Oxford Instrument plc, Abingdon, UK) dispersion X-ray (EDX) equipped with FESEM.

The thermal stability of fillers and membranes were studied by using Thermo plus EVO2 instrument (Rigaku, Tokyo, JPN). The membrane samples were heated in the air up to 750 °C at a heating rate 10 °C/min. Fourier Transform Infra-Red (FTIR) was used to determine the chemical functional group of ZIF-8 and amine-functionalized ZIF-8 particles by using Perkin Elmer One spectrometer (Perkin Elmer Inc., Waltham, USA). The ZIF-8 filler in powder form was compressed into pellet by blending with KBr. The analysis was then conducted in a transmittance mode from 400 to 4000 cm^−1^ using 50 scans at room temperature. Meanwhile, the functional groups in the membranes were investigated by using FTIR spectroscopy (Perkin Elmer Inc., Waltham, USA) operated in the attenuated total reflection mode (ATR) equipped with a diamond crystal. The membrane films were scanned 50 times for each sample and the scanning was from 650 to 4000 cm^−1^ at room temperature.

On the other hand, the fractional free volume (FFV) of the membranes was calculated according to Equations (1)–(4) as follows [30,32,53]:(1)$$FFV=\frac{V-V_{0}}{V}$$
(2)$$V=\frac{M}{\rho }$$
(3)$$\rho =\frac{W_{air}}{W_{air}-W_{liquid}}\rho_{0}$$
(4)$$V_{0}=1.3\times V_{vdw}$$
where *FFV* denotes the fractional free volume, *V* is the molar volume of the repeating unit of polymer (cm^3^/mol), *V_o_* refers to the volume occupied by polymer chains (cm^3^/mol) and *V_vdw_* represents the Van der Waals molar volume (cm^3^/mol) which is calculated from the group contribution method of Bondi (1964). *M* refers to the molecular weight of the repeating unit of polymer (g/mol), *ρ* is the density of the membrane (g/cm^3^), *ρ*_0_ is the density of the auxiliary liquid (g/cm^3^), *W_air_* and *W_liquid_* represent the membrane weight measured in the air and liquid (g), respectively. The density of membrane films was measured using an electronic balance (Mettler Toledo, OHAUS CP224C, OHAUS Intruments, Kuala Lumpur, Malaysia) equipped with a density determination kit via Archimedean principle method. In the measurement, high purity ethanol was used as the auxiliary liquid.

Meanwhile, the CHN compositions of the resultant amine functionalized filler were determined by CHNS elemental analysis using vario MACRO cube (Elementar Analysensysteme GmbH, Langenselbold, DEU). The results are shown in Appendix A. Referring to Appendix A, the nitrogen and carbon elements increased after the functionalization indicated that the amine functional groups are successfully attached on the surface of ZIF-8.

### 2.6. Gas Permeation Measurements

The permeability of pure CO_2_ and CH_4_ gases were measured over the resultant membranes at room temperature and 3.5 bar by constant pressure technique using custom built gas permeation test rig. The details of the permeation equipment were explained elsewhere [54]. The membrane sample was mounted into the membrane test cell, then vacuumed for overnight before performance test to remove moistures and trapped gas. After that, the upstream gas was fed into test cell and the flow rate of the permeate streams was measured using bubble flow meter. Permeability of CO_2_ and CH_4_ were calculated using Equation (5) as follows [30].
(5)$$P=\frac{V_{P}t}{A_{m}\left(p_{h}-p_{l}\right)}$$
where *V_p_* is the permeate flow rate (cm^3^(STP)/s)_,_
*t* is the membrane thickness (cm), *A_m_* is the membrane area (cm^2^), *p_h_* and *p_l_* are the pressure in feed and permeate side, respectively (cmHg). The permeability of the membrane is defined in unit Barrer. The ideal selectivity of membranes was calculated by using ratio of permeability for two gases using Equation (6) as follows [30]:(6)$$\alpha_{CO_{2}/CH_{4}}=\frac{P_{CO_{2}}}{P_{CH_{4}}}$$
where $\alpha_{CO_{2}/CH_{4}}$ indicates the ideal selectivity of membrane. In addition, the contribution of filler (µ) in enhancing the CO_2_ gas permeability can be calculated using Equation (7) as follows [24]:(7)$$\mu_{CO_{2}}=\frac{P_{CO_{2}}^{CM}-P_{CO_{2}}^{P}}{P_{CO_{2}}^{P}}\times 100$$
where $\mu_{CO_{2}}$ is the contribution of filler for enhancement of CO_2_ gas permeability, $P_{CO_{2}}^{CM}$ and $P_{CO_{2}}^{P}$ are the permeabilities in mixed matrix membrane and pure membrane, respectively. Three samples of each membrane have been tested to evaluate the reproducibility of the experimental results and the error of the data obtained is within ±3%.

## 3. Results and Discussion

### 3.1. Characterization of Fillers and Membranes

#### 3.1.1. X-Ray Diffraction (XRD)

Figure 1 shows the XRD pattern of non-functionalized and amine-functionalized ZIF-8 fillers. Referring to Figure 1, the major peaks of the prepared fillers are found at 2θ = 7.50°, 10.50°, 12.80°, 16.20°, 18.30° and 23.50°. The peak that appears at 2θ = 7.50° are related to X-ray reflection from the plane (110) of ZIF-8 crystalline structure and is used as a reference of the ZIF-8 identification [37]. These patterns are in good agreement with the reported literature [44,45,50], which confirmed the phase purity of ZIF-8. Besides, it can be seen that the incorporation of amine-functional groups did not affect the structure and crystallinity of ZIF-8.

Figure 2 shows the XRD patterns of the resultant membranes. Referring to Figure 2, the peak at 23.5° corresponded to ZIF-8 structure was found for all the resultant membranes (M1-M6), which confirmed the presence of amine-functionalized ZIF-8 fillers in the mixed matrix membranes. In addition, no major peaks can be seen in pure 6FDA-durene membrane (M0). This confirmed the amorphous nature of the polymer, which matches well with the reported literature [55].

#### 3.1.2. Field Emission Scanning Electron Microscopy (FESEM) and Energy Dispersive X-Ray (EDX) Spectroscopy Mapping

Figure 3 presents the FESEM images of the pure and mixed matrix membranes fabricated in this work. It can be seen from Figure 3 that amine-functionalized ZIF-8 fillers are encapsulated in the polymer phase (M1-M6). Besides, the presence of concentric cavities in membranes M2, M3, M5 and M6 demonstrated strong interfacial contact between amine-functionalized ZIF-8 fillers and 6FDA-durene polymer [37,41]. However, membranes incorporated with APTMS-ZIF-8 fillers (M1 and M4) showed different morphology compared to the other membranes incorporated with AAPTMS-ZIF8 fillers (M2 and M5) and AEPTMS-ZIF-8 fillers (M3 and M4). Apart from that, a cluster of APTMS-ZIF-8 fillers can be seen in the polymer phase. This might be due to the self-condensation between aminopropyl-silane molecules, which may form a cluster on the ZIF-8 filler surface [56].

Figure 4 displays the EDX mapping analysis of the resultant membranes. Based on Figure 4, the presence of amine-functionalized ZIF-8 fillers was confirmed with the presence of Zn element which uniformly distributed throughout the resultant membranes. In addition, the presence of Zn element in the M4, M5 and M6 is more compact compared to M1, M2 and M3 due to the increase of amine-functionalized ZIF-8 fillers loadings from 0.5 wt% to 1.0 wt%. The atomic (%) composition of Zn element obtained from the EDX analysis for the membranes loaded with 0.5 wt% and 1.0 wt% of functionalized fillers was about 0.11 and 0.45, respectively (Appendix A).

#### 3.1.3. Thermogravimetric Analysis (TGA)

Figure 5 shows the thermal stability of the membranes. It can be seen from Figure 5 that all the resultant membranes demonstrated first weight loss started at 100 °C due to the moisture and residual solvent removal from the resultant membranes. In addition, it can be observed that the decomposition temperature of the pure 6FDA-durene membrane begins at 420 °C, in accordance with the reported literature [57]. The weight loss of 10% at 472 °C found in 6FDA-durene membrane (M0) is mainly due to polymer degradation at initial stage [57]. Meanwhile, the major weight loss occurred between 500 °C and 600 °C in all membranes (M0-M6) due to the decomposition of 6FDA-durene polymer. It has been found that the incorporation of amine-functionalized ZIF-8 fillers into 6FDA-durene polymer slightly reduced the thermal stability of the resultant mixed matrix membranes (M1-M6). This could be due to the disruption of polymer chain resulting from the addition of the functionalized filler which cause the mobility of polymer chain and decomposed at lower thermal energy [53]. Moreover, amine functional group decomposed at lower temperature [24] which indirectly reduces the thermal stability of the membranes.

It can be observed from Figure 5 that the thermal behavior of the mixed matrix membranes is not significantly impacted by the incorporation ZIF-8 fillers functionalized with different amine groups as compared to the pure 6FDA-durene membrane. Meanwhile, the difference in thermal stability and weight loss of mixed matrix membranes incorporated with 0.5 wt% and 1 wt% was not significant mainly due to small different in filler loading of the membranes. This finding is supported by the reported literature, which incorporating low loading of ZIF-8 fillers in the polymer matrix and suggested that ZIF-8 loading of <5 wt% of the total solids did not significantly affect the overall thermal stability of the prepared membranes [44].

#### 3.1.4. Attenuated Total Reflectance—Fourier Transform Infrared (ATR-FTIR)

Figure 6 shows the FTIR spectra of the ZIF-8 and amine-functionalized ZIF-8 fillers. Referring to Figure 6, the peak at 419 cm^−1^ attributed to the Zn-N stretching [53] while absorption band at 1580 cm^−1^ is corresponding to C=N stretch [50]. In addition, the characteristics peaks between 900 cm^−1^ to 1450 cm^−1^ are ascribed to the in-plane bending of the imidazole ring, while the peak at 693 cm^−1^ and 758 cm^−1^ belong to the out-of-plane bending of the rings [53,58]. Besides, the existence of characteristic peaks at 1382 cm^−1^ and 1515 cm^−1^ are corresponding to the aromatic stretching [53]. The absorption bands at 2928 cm^−1^ and 3137 cm^−1^ are detected for aromatic and aliphatic C–H stretch of the imidazole [50,58]. While, the characteristic peaks at 3510 cm^−1^ are observed at amine-functionalized ZIF-8 fillers including APTMS-ZIF-8, AAPTMS-ZIF-8 and AEPTMS-ZIF-8 which belong to the asymmetric amine group stretching bonds [49], thus confirmed the presence of amine functional group in the ZIF-8 fillers.

Figure 7 displays the ATR-FTIR of the membranes. Referring to Figure 7, 6FDA-durene membrane (M0) consist of the characteristic peaks at 718 cm^−1^, 1352 cm^−1^, 1715 cm^−1^ and 1785 cm^−1^. The existence of characteristic peaks at 718 cm^−1^ is attributed to deformation of imide ring which confirmed the formation of imide group. The absorption bands at 1715 cm^−1^ and 1785 cm^−1^ are corresponding to C=O symmetric and asymmetric stretching [41,59]. Meanwhile, peaks at 1352 cm^−1^ and 1250 cm^−1^ found in pure 6FDA-durene membrane (M0) are the C–N stretching of the imide group and C–F stretch in CF_3_ group [50]. The peaks at wavenumber of 1420 cm^−1^ are attributed to C=N bonds. Compared to the pure 6FDA-durene membrane (M0), it can be seen that all the membranes demonstrated similar peaks (M1-M6). This could be due to the small loadings of amine-functionalized fillers (APTMS-ZIF-8, AAPTMS-ZIF-8 and AEPTMS-ZIF-8) in the range of 0.5 wt% to 1.0 wt% incorporated into the polymer matrix. Therefore, the peaks corresponding to the amine functionalized fillers are hardly identified.

#### 3.1.5. Free Fractional Volume (FFV) Analysis

Table 2 shows the density and fractional free volume (FFV) values of the resultant membranes. Referring to Table 2, pure 6FDA-durene membrane (M0) showed the highest density of 1.440 g/cm^3^ and lowest FFV of 0.1981 compared to the mixed matrix membranes (M1-M6). The FFV of the 6FDA-durene membrane is comparable to the previously reported value [60]. On the other hand, the FFV and the density values of the M1 to M6 are in the range of 0.1981 to 0.2024 and 1.432 to 1.440 g/cm^3^, respectively. It can be observed that the membranes showed increment of FFV and reduction of density compared to pure 6FDA-durene membrane. The increment of FFV and the reduction of the density values of the mixed matrix membranes could be due to the incorporation of amine functionalized ZIF-8 fillers into 6FDA-durene polymer which increase the free volume cavities and cause the distraction of polymer chain packing [61], and thus resulted in the increment of polymer interchain gap [57].

It can also be observed that membranes incorporated with AAPTMS-ZIF-8 (M2) and AEPTMS-ZIF-8 (M3) showed the highest increment of FFV about 2.2% and 1.9% compared to pure 6FDA-durene membrane (M0). The increment might be due to the ability of diamine and triamine groups to improve the free volume cavities by optimizing the interruption of polymer chain packing [30]. The mixed matrix membrane incorporated with APTMS-ZIF-8 (M1) which functionalized with monoamine group showed a lower increment compared to mixed matrix membranes containing ZIF-8 fillers functionalized with diamine and triamine groups. This might be due to the blockage of free volume cavities resulting from additional free volume in polymer matrix [30]. However, the increment of functionalized ZIF-8 fillers loadings from 0.5 wt% to 1.0 wt% showed a reduction of FFV for M4, M5 and M6. The reduction of FFV might be due to the compactness of the polymer chain packing [62] resulting from excessive amount of functionalized ZIF-8 fillers in the polymer phase.

### 3.2. Gas Permeation Results

The pure gas permeation performance of the membranes was tested by using gas permeation rig at fixed feed pressure of 3.5 bar and room temperature. Table 3 summarized gas permeability, CO_2_/CH_4_ selectivity and **µ_CO_2__** of all the membranes fabricated in this work. Referring to Table 3, the CO_2_ permeability and CO_2_/CH_4_ selectivity increase while CH_4_ permeability decrease for all the resultant mixed matrix membranes compared to pure 6FDA-durene membrane. This could be due to the introduction of amine-functionalized ZIF-8 fillers which increased the inter-segmental spacing between polymer chain near the fillers [33,63]. In addition, it can be observed that the values of **µ_CO_2__** are positive as shown in Table 3, which agree that the incorporation of amine-functionalized ZIF-8 filler enhanced the gas permeability [24]. Besides, the formation of porous network, the increment of the free volume in 6FDA-durene polymer matrix also contributed to the improvement of CO_2_ permeability. Meanwhile, the pure 6FDA-durene membrane showed CO_2_ permeability of 510.3 Barrer and CO_2_/CH_4_ selectivity of 8.6 which is comparable with the reported literature [51].

Besides, it can be observed that functionalization of ZIF-8 fillers with different type of amine groups including monoamine (APTMS), diamine (AAPTMS) and triamine (AEPTMS) group resulted in an improvement of CO_2_ permeability and CO_2_/CH_4_ selectivity in the range of 1.6–61.7% and 56.5–224.0%, respectively. The highest CO_2_ permeability of 825.1 Barrer and CO_2_/CH_4_ selectivity of 26.2 was obtained from membrane M2. This could be because of the presence of AAPTMS on ZIF-8 fillers surface reduced the degree of partial blockage of ZIF-8 fillers by introducing a space between polymer chain and ZIF-8 surface [14]. Besides, the functionalization of ZIF-8 fillers with AAPTMS also contributed to a better dispersion of the fillers in the polymer matrix and provide a better absorption and transportation of gases, resulting better separation performance [33]. Apart from that, the increment of free volume from 0.2009 (M1) to 0.2024 (M2) could also improve the diffusion properties and enhanced the CO_2_ permeability.

On the other hand, membrane loaded with 0.5 wt% AEPTMS-ZIF-8 (M3) exhibited CO_2_ permeability of 713.8 Barrer and CO_2_/CH_4_ selectivity of 27.9. It was found that reduction of CO_2_ permeability (13.5%) and increment of CO_2_/CH_4_ selectivity (6.3%) were observed on M3 (0.5 wt% AEPTMS-ZIF-8/6FDA-durene membrane) compared to M2 (0.5 wt% AAPTMS -ZIF-8/6FDA-durene membrane). This result could be due to the rigidified polymer area near the fillers which improved the diffusivity due to less mobility of polymer chains [14].

CO_2_ permeability and CO_2_/CH_4_ selectivity showed a different trend for the membrane loaded with 0.5 wt% APTMS-ZIF-8 (M1) compared to the membranes loaded with 0.5 wt% AAPTMS-ZIF-8 (M2) and 0.5 wt% AEPTMS-ZIF-8 (M3). The CO_2_ permeability of 825.1,713.8 Barrer, 649.6 Barrer and CO_2_/CH_4_ selectivity of 26.2, 27.9, 17.4 were obtained for M2, M3 and M1, respectively. The lower CO_2_ permeability and CO_2_/CH_4_ selectivity of M1 could be because of the pore blockage of APTMS-ZIF-8 fillers due to higher number of alkoxysilane on the ZIF-8 filler surface [32]. Furthermore, the reduction of FFV value from 0.2024 (M2) and 0.2020 (M3) to 0.2009 (M1) also support the decrease in CO_2_ permeability due to the disruption of polymer chains by the presence the APTMS-ZIF-8 fillers, leading to the decrease in free volume and chain mobility [64].

On the other hand, the increment of amine-functionalized ZIF-8 fillers loading from 0.5 wt% to 1.0 wt% in the polymer matrix resulted in a reduction of gas permeation performance of M4-M6 compared to M1-M3. However, the gas permeation performance still higher compared to pure 6FDA-durene membrane (M0). This could be due to the additional reaction occurred between polyimide chain and amine-functional group because of the increment of amine-functionalized ZIF-8 fillers loading and thus, resulted in polymer rigidication. Meanwhile, the reduction of CO_2_/CH_4_ selectivity for the membranes loaded with 1.0 wt% could be due to the filler pore blockage by the polymer [44]. In addition, the decreased of FFV value for the membranes loaded with 1.0 wt% of amine-functionalized ZIF-8 fillers also caused the decrease in the diffusivity of the gas molecules and led to the reduction of gas permeation performance [65]. Apart from that, the CO_2_ permeability and CO_2_/CH_4_ selectivity obtained from mixed matrix membranes loaded with 0.5 wt% amine-functionalized ZIF-8 fillers is higher than that of the performance of 6FDA-durene membrane loaded with 5.0 wt% ZIF-8 fillers reported in previous work [50]. The results obtained in this work show that the gas permeation performance of the mixed matrix membrane can be enhanced even at low loading of amine-functionalized ZIF-8 filler, attributed to strong CO_2_ affinity and better dispersion of amine-functional groups in the polymer matrix.

Overall, the incorporation of different amine-functionalized ZIF-8 fillers into 6FDA-durene polymer resulted in higher CO_2_ permeability and CO_2_/CH_4_ selectivity compared to pure 6FDA-durene membrane. This could be due to higher CO_2_ solubility and more free volume availability in the membrane after incorporation of fillers [33]. Besides, the functionalization of ZIF-8 filler by APTMS, AAPTMS and AEPTMS also induced better dispersion of fillers and having better affinity for CO_2_ mainly due to the present of amine groups. In this work, functionalization with diamine group showed more significant improvement CO_2_ permeability and CO_2_/CH_4_ selectivity due to the steric hindrance effect compared with monoamine and triamine groups [30,66,67].

### 3.3. Comparison with Robeson Upper Bound

Figure 8 shows the comparison of gas permeation performance obtained in this work with Robeson 2008 upper bound. It can be observed that 6FDA-durene membrane (M0) lies below 1991 Robeson upper bound limit. On the other hand, mixed matrix membranes loaded with 0.5 wt% AAPTMS-ZIF-8 and AEPTMS-ZIF-8 (M2 and M3) successfully lie on 2008 Robeson upper bound limit. Meanwhile, mixed matrix membranes containing 0.5 wt% APTMS-ZIF-8 fillers (M1) and 1.0 wt% AAPTMS-ZIF-8 (M5) fillers are lie on 1991 Robeson upper bound. Besides, mixed matrix membranes containing 1.0 wt% of APTMS -ZIF-8 and AEPTMS -ZIF-8 (M4 and M6) are closer to the 1991 Robeson upper bound. It can be also found that the incorporation of amine-functionalized ZIF-8 fillers including APTMS-ZIF-8, AAPTMS-ZIF-8 and AEPTMS-ZIF-8 successfully improved the CO_2_ permeability and CO_2_/CH_4_ selectivity. Thus, it can be concluded that the functionalization of fillers with suitable amino-functional group could further improve the separation performance of the membrane by enhancing the interfacial properties between the filler and polymer matrix.

## 4. Conclusions

Mixed matrix membranes comprising amine-functionalized ZIF-8 were fabricated successfully. XRD results indicated major peaks at 7.50°, 10.50°, 12.80°, 16.20° and 18.30° were in agreement with reported literature. FESEM and EDX mapping displayed that the fillers were homogenously dispersed in the polymer phase. From FFV analysis, the incorporation of amine-functionalized ZIF-8 lead to the increment of FFV values ranged from 0.1981 to 0.2024. Mixed matrix membranes loaded with 0.5 wt% of AAPTMS-ZIF-8 and AEPTMS-ZIF-8 displayed highest CO_2_ permeability of 825.1 Barrer and 713.8 Barrer, with CO_2_/CH_4_ selectivity of 26.2 and 27.9, respectively. Both membranes successfully lie on 2008 Robeson’s upper bound trade-off limit. In conclusion, the functionalization of filler by using different amino-functional groups has potential in enhancing CO_2_ permeability and CO_2_/CH_4_ selectivity compared to the pure 6FDA-durene membrane. Thus, proper selection of amino-functional group for filler’s functionalization and optimum loading of filler are crucial in order to ensure the functionalization will contribute to the improvement of the membrane performance.

## Figures and Tables

**Figure 1 polymers-11-02042-f001:**
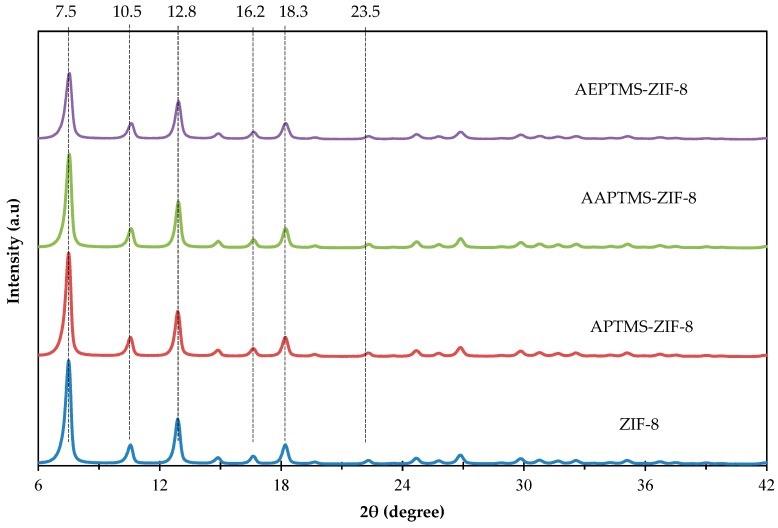
X-ray diffraction (XRD) patterns of ZIF-8 and amine-functionalized ZIF-8 fillers.

**Figure 2 polymers-11-02042-f002:**
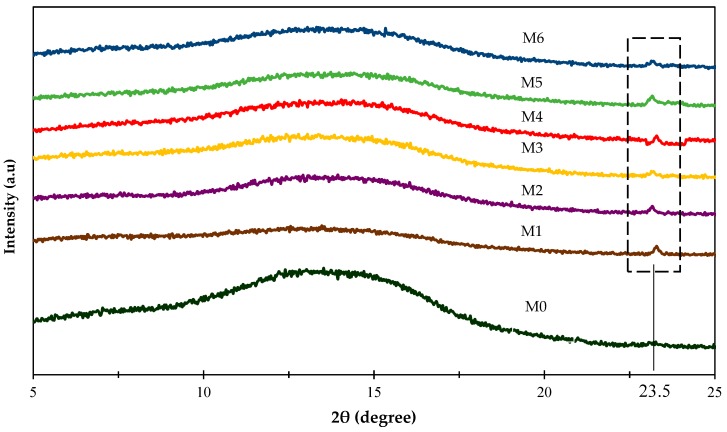
X-ray diffraction (XRD) patterns of the resultant membranes.

**Figure 3 polymers-11-02042-f003:**
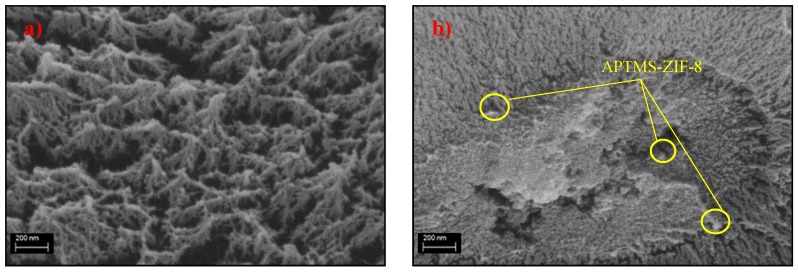
Field emission scanning electron microscope (FESEM) images of membranes (**a**) M0, (**b**) M1; (**c**) M2; (**d**) M3; (**e**) M4; (**f**) M5 and (**g**) M6.

**Figure 4 polymers-11-02042-f004:**
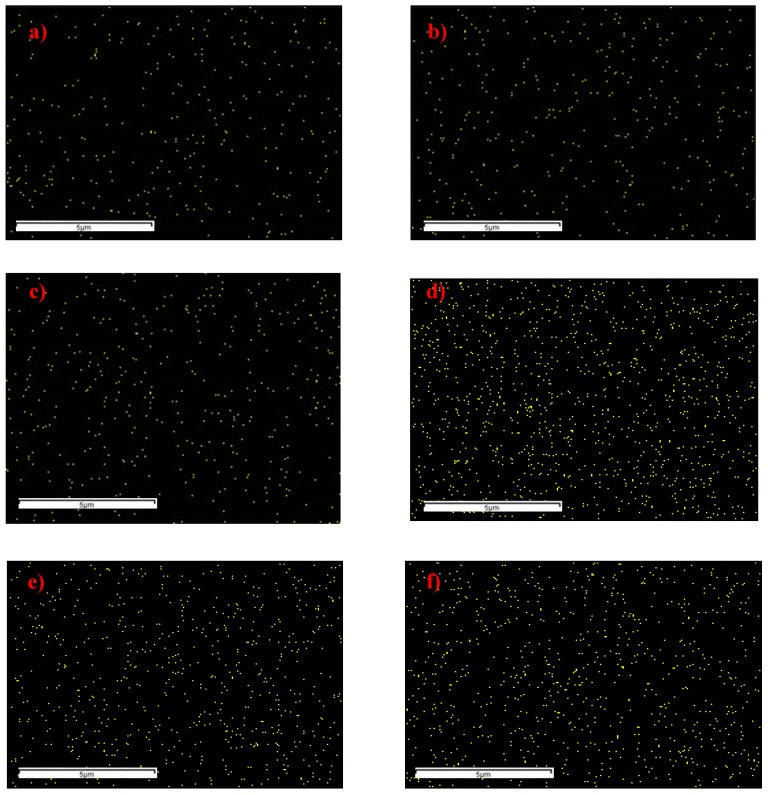
Energy dispersive X-ray (EDX) mapping analysis of the resultant mixed matrix membranes (**a**) M1; (**b**) M2; (**c**) M3; (**d**) M4; (**e**) M5 and (f) M6.

**Figure 5 polymers-11-02042-f005:**
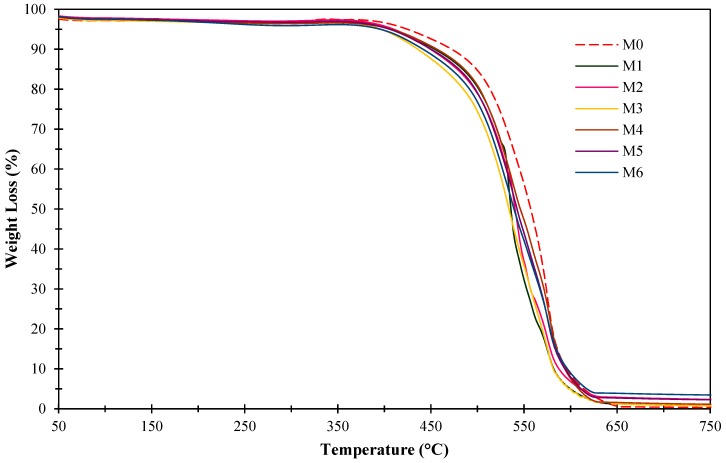
Thermogravimetric analysis (TGA) curves of the pure 6FDA-durene membrane (M0) and mixed matrix membranes (M1-M6).

**Figure 6 polymers-11-02042-f006:**
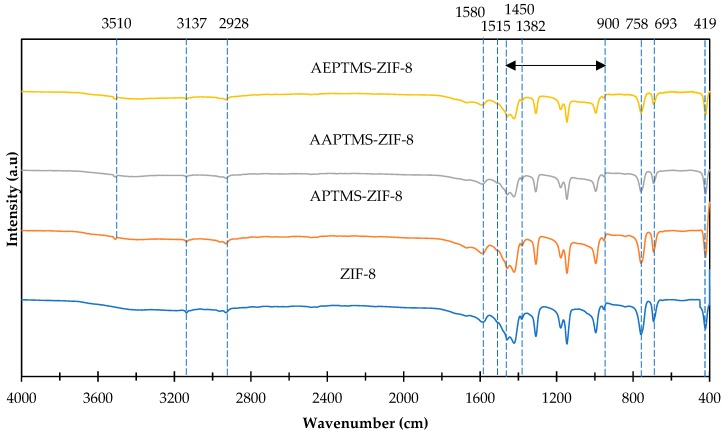
Fourier transform infrared (FTIR) spectra of ZIF-8 and amine-functionalized ZIF-8 fillers.

**Figure 7 polymers-11-02042-f007:**
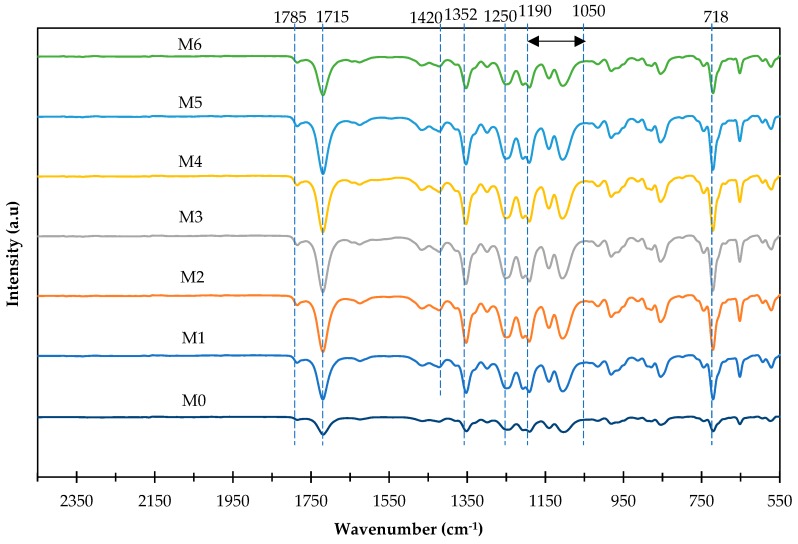
Attenuated total reflection- Fourier transform infrared (ATR-FTIR) spectra of the resultant membranes.

**Figure 8 polymers-11-02042-f008:**
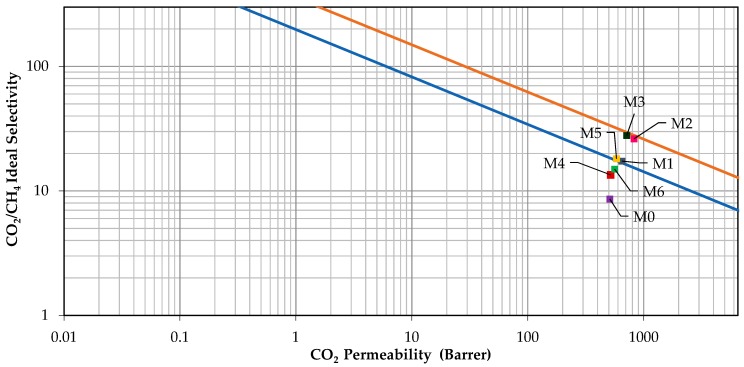
Comparison of gas separation performance of the membranes fabricated in this work with Robeson’s upper bound.

**Table 1 polymers-11-02042-t001:** Summary of the membranes fabricated in this work.

Sample	Synonym	6FDA-durene (wt%)	APTMS-ZIF-8 (wt%)	AAPTMS-ZIF-8 (wt%)	AEPTMS-ZIF-8 (wt%)
6FDA-Durene	M0	100.0	-	-	-
0.5 wt% APTMS-ZIF-8 /6FDA-Durene	M1	99.5	0.5	-	-
0.5 wt% AAPTMS-ZIF-8 /6FDA-Durene	M2	99.5	-	0.5	-
0.5 wt% AEPTMS-ZIF-8 /6FDA-Durene	M3	99.5	-	-	0.5
1.0 wt% APTMS-ZIF-8 /6FDA-Durene	M4	99.0	1.0	-	-
1.0 wt% AAPTMS-ZIF-8 /6FDA-Durene	M5	99.0	-	1.0	-
1.0 wt% AEPTMS-ZIF-8 /6FDA-Durene	M6	99.0	-	-	1.0

**Table 2 polymers-11-02042-t002:** Density and fractional free volume (FFV) of the membranes.

Membrane	Density (g/cm^3^)	Fractional Free Volume (FFV)
M0	1.440	0.1981
M1	1.435	0.2009
M2	1.432	0.2024
M3	1.433	0.2020
M4	1.439	0.1987
M5	1.436	0.2003
M6	1.439	0.1983

**Table 3 polymers-11-02042-t003:** Gas permeabilities, CO_2_/CH_4_ selectivities and **µ_CO_2__** of the membranes fabricated in this work.

Membrane	Loading (wt. %)	P_CO_2__ (Barrer)	P_CH_4__ (Barrer)	CO_2_/CH_4_ Selectivity	µ_CO_2__ (%)
M0	0	510.3	59.34	8.60	-
M1	0.5	649.61	37.45	17.35	27.30
M2	0.5	825.14	31.48	26.21	61.70
M3	0.5	713.79	25.62	27.86	39.88
M4	1.0	518.33	38.50	13.46	1.57
M5	1.0	582.54	32.10	18.15	14.16
M6	1.0	561.56	37.55	14.96	10.04

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
