# Peer review of "Tailoring CO2/CH4 Separation Performance of Mixed Matrix Membranes by Using ZIF-8 Particles Functionalized with Different Amine Groups"

_polymers, 2019, doi:10.3390/polym11122042_

Round 1

Reviewer 1 Report

The manuscript by Nadia Hartini Suhaimi et al. introduces a comparative study in regards to the CO2/CH4 separation performance of the membranes fabricated from blending amine-functionalized ZIF-8 particles into a polymer matrix. The amine species and filler fraction are varied. The authors aim to correlate the amine species with the gas separation performance of the resultant membrane. However, there is insufficient evidence to support this connection. Moreover, there are some conceptual and characterization issues that need to be addressed. Specifically:

The degree of the amine functionalization in each ZIF-8 sample is unclear, which makes the comparison of membrane performance unreliable. For example, can the better performance of M2 induced by a higher content of the amine group on the particle surface? To address this, I recommend using TGA and/or element analysis on each amino-ZIF-8 particles to quantitively specify the amine content in each particle. In addition, the composite of the membrane should list wt% for each component (ZIF-8, amine, matrix). The trend of filler loading is unreasonable. In TGA, some 0.5% loading membrane has a higher residual weight than that of the 1% loading membranes. In addition, all 1% membranes have lower FFV and lower gas separation performance compared with the 0.5% loading ones. This does not make sense to the authors’ conclusion. All these results suggest the “effective” loading in 1% membranes is lower. Maybe the filler particles precipitated on the bottom and were left when peeling off the membrane? The authors should figure out the cause of this abnormal trend and state the reproducibility of the experiment. The authors use the term “composite membrane” to refer to their membrane, however, the “mixed-matrix membrane” is a more proper term in membrane community to avoid confusing with the “thin film composite membrane”. The calculation of FFV do not consider amine on the surface of ZIF-8. If the amine content in the filler is fairly high, the volume of amine should be taken into consideration. Reference [55] was cited to explain the abnormal trend in FFV. This reference only shows the correlation between inter chain distance and the FFV, but why higher filler loading can reduce chain distance is not mentioned. Again, the authors should try to figure out this abnormal trend.

Author Response

Dear Reviewer,

Kindly see the attachment.

Thank You

Reviewer 2 Report

In this work, the authors modified ZIF-8 with amine functional groups to enhance interaction at the polymer/filler interface, leading to improved performances in the mixed-matrix membranes for CO2/CH4 separation. While there is a lack in the overall novelty (as also pointed out by the authors in the introduction), the work is comprehensively studied and well-written. I would like to add some comments which I hope can help to improve the overall quality of the manuscript.

1. Introduction: Generally, as the authors have pointed out, there are many works already published on functionalized ZIF-8 for mixed-matrix membranes. I feel that in the introduction there is a need to justify stronger the motivation behind this work, besides just stating the obvious advantages of using MOF fillers and functionalizing them.

2. Figure 3: It is not clear to me that the circled areas consist of ZIF-8 fillers. Did the authors characterized the particle size distribution of the ZIF-8 fillers alone? Typically, ZIF-8 crystals are much bigger than what was shown. So the question is whether they are really ZIF-8 or the integrity of your ZIF-8 crystals were compromised leading to a smaller crystal size.

3. Figure 3: Again it is not apparent to me that there are unselective voids between the so-called ZIF-8 and matrix. Figure 3e happened to be taken at an area where there are some tearing in the polymer matrix which could have resulted in the voids appearing. 

4. Figure 4: On the same note, did the authors scan the EDX based on the SEM images in Figure 3? If the circled areas indeed comprise ZIF-8 crystals, then the EDX scans should not appear this way. Alternatively, the EDX elemental mappings seem to show very low signals which could be just noise? What is the % composition of Zn obtained from the EDX analysis?

5. Figure 6: It appears that the extent of functionalization is not great as demonstrated by the weak NH2 stretching bands? Did the authors carry out characterization to confirm the successful synthesis of functionalizaed ZIF-8?

6. Table 3. There is no control carried out in this work. The authors did not measure the performance of the membranes with unmodified ZIF-8 fillers. So I am unsure if the performance enhancement is due to the modification or the filler itself.

7. There is no need for Figures 8 and 9 as the results were summarized in Table 3. 

8. To elevate the discussion and to also inject novelty to this piece of work, I suggest that we can discuss effectiveness from the filler materials perspective. Recently, our group initiated a filler enhancement index, the Findex, for CO2/CH4 MMMs separation, which i think is useful for this work. The authors should add in the results for the unmodified ZIF-8 and then compare the change in the values of Findex. I anticipate that the Findex values of the modified ZIF-8 will be higher than the unmodified ZIF-8, and thus opening up discussion of greater depth. I recommend the authors to read our paper on Chem. Rev. 2018 118 (18) 8655-8769 for more information on the Findex.

Author Response

(The authors gave the same response as above.)

Round 2

Reviewer 1 Report

Dear Editor, the modifications by the authors seem reasonable, and this manuscript can be published in the current form.

Author Response

Dear Reviewer,

Thank you very much for the reviewer's comments. The comments and suggestions are greatly appreciated. We have revised the manuscript in line with the comments. We are enclosing a copy of the changes made (highlighted in yellow) in the revised manuscript in view of there viewer's comments.

We are herewith submitting a copy of our revised manuscript for your kind attention.

We hope that you will find the revised manuscript in order and suitable for publication. Thank you once again for your support and a great help.

Thank You

Reviewer 2 Report

The authors have adequately addressed most of my concerns and I would like to just point out some minor points before accepting the manuscript for publication:

Introduction line-100: Authors' claim on different amine groups used in this work include primary, secondary and tertiary amine is incorrect. AEPTMS is not a tertiary amine but a triamine. Figure S2: Can the authors also add in the elemental atomic (%) of M0 as a reference. Line-182: Please change the symbol of density to ρ. I think there is either a mistake in entry or an error during PDF conversion.  Line 333-334: It appears puzzling to me that the fillers at 40 nm particle size can occupy the free volume inside the polymer matrix. I would think that excessive filler loading at 1 wt% changes the polymer packing to a greater extent that it reduces the FFV is a more plausible reason. Line 378-385: The authors attempted to use FFV and polymer rigidification to rationalize the drop in CO2 permeability but this cannot explain for the decrease in the selectivity as compared to MMMs loaded with 0.5 wt% fillers. Please kindly comment on why the drop in selectivity.  

Author Response

(The authors gave the same response as above.)
